# Perish the thawed? EDTA reduces DNA degradation during extraction from frozen tissue

Ella Messner[1,2⊕], Lev Becker[1,3⊕], Mia L. DeSanctis[1,4], Elizabeth A. Soranno[1,5], Ryan Pianka[1], Caileigh Pierce[1,6], Molly Johnson[1], Rosalia Falco Poulin[1], Hannah J. Appiah-Madson[1], Daniel L. Distel[1]*

1 Ocean Genome Legacy Center, Marine and Environmental Sciences, Northeastern University, Nahant, Massachusetts, United States of America, 2 The Pennsylvania State University, University Park, Pennsylvania, United States of America, 3 University of Connecticut, Storrs, Connecticut, United States of America, 4 New England Biolabs, Ipswich, Massachusetts, United States of America, 5 Commercial Fisheries Research Foundation, South Kingstown, Rhode Island, United States of America, 6 Harvard Medical School, Boston, Massachusetts, United States of America

⊕ These authors contributed equally to this work.
* d.distel@northeastern.edu (DLD)

## Abstract

Cryopreservation is the gold standard for preserving high molecular weight (HMW) DNA (>10 kb) in tissue samples. However, frozen tissues are typically thawed either before or during DNA extraction, which can lead to substantial DNA degradation. In this study, we thawed the previously frozen tissues of 10 marine species (five fishes and five invertebrates) in the preservatives EDTA (250 mM, pH 10) or ethanol (EtOH, 95%) and maintained them in their respective preservatives overnight at 4°C before DNA extraction. We then compared the recovery of HMW DNA in these extracts to extracts prepared directly from frozen tissues. To evaluate the effect of these treatments on HMW DNA recovery, we determined the percentage of high molecular weight DNA (%HMW) and yield of HMW DNA normalized by tissue weight (nY) in each DNA extract. The average %HMW values for eight of the 10 species and the average nY values for five of the 10 species were significantly higher in extracts from EDTA-treated tissues compared to extracts from untreated frozen tissues. For all 10 species, we observed no significant decreases in average %HMW or nY values in extracts of EDTA-thawed tissues compared to those extracted directly from frozen tissues. In contrast, EtOH treatment did not significantly improve the average %HMW or nY values in extracts from tissues of nine of the 10 species when compared to extracts prepared directly from frozen tissues. Therefore, investigators may consider EDTA treatment as a simple method for improving HMW DNA recovery from frozen tissues.

## Introduction

Cryopreservation at ultracold temperatures, typically -80 to -196°C, is the preferred method for preserving DNA in tissue samples for genetic or genomic research [1–6]. For example, a

**Data availability statement:** All relevant data are within the manuscript and its Supporting Information files except sequence data, which are deposited to NCBI Nucleotide Database under the accession numbers listed in S1 Table.

**Funding:** This work was funded by a grant from Cell Signaling Technologies Inc. Annual Research Grant Program ( to DLD) and received support from the Donald Comb Research Scholarship Fund (to DLD), the Ocean Genome Legacy Operations Fund (to DLD), and the Francis Goelet Charitable Lead Trust (to DLD). Resources purchased with funds from the National Science Foundation's Field Stations and Marine Laboratories program (DBI 1722553 to Northeastern University) were used to generate data for this manuscript. The funders had no role in study design, data collection and analysis, decision to publish, or manuscript preparation.

**Competing interests:** The authors have declared that no competing interests exist.

2014 survey of 45 genomic resource collections associated with natural history museums and research institutions found that 91% maintained frozen materials, 78% maintained ultracold freezers, and 27% maintained liquid nitrogen dewars [7]. Low temperatures reduce DNA degradation by inhibiting chemical and biological activity, including nuclease activity that can rapidly reduce the average molecular weight of DNA in tissues [5,8]. Such fragmentation renders DNA less suitable for many downstream applications, including long-read DNA sequencing, reduced-representation sequencing, whole genome and metagenome sequence assembly, and preparation of large insert libraries [1,3,9,10].

While freezing may protect against DNA degradation, this protection is lost when the materials are thawed [1,11]. Because most DNA extraction procedures occur in liquid media, tissues must be thawed, at least briefly, either before or during extraction, which potentially allows DNA degradation to occur. To minimize DNA damage due to thawing, researchers often grind frozen tissues in liquid nitrogen or on dry ice before transferring the frozen material to an appropriate DNA extraction medium [12–14]. While effective, these methods are technically demanding, inconvenient, dangerous, and prone to accidental thawing and tissue loss during transfer, especially for small samples.

When applied to fresh tissues, chemical preservatives can substantially reduce DNA degradation during subsequent handling and storage [8]. In this investigation, we ask whether the same is true when applied to frozen tissues. Specifically, we ask whether placing frozen tissues in the chemical preservatives ethylenediaminetetraacetic acid (EDTA) or ethanol (EtOH) during the transition from the frozen to the thawed state can provide simple, convenient, safe, inexpensive, and effective methods for reducing DNA degradation during subsequent handling and DNA extraction.

We chose to evaluate EtOH as it is the most commonly used tissue preservative compatible with DNA preservation. It is widely employed for field collection and long-term storage of biological materials, particularly in natural history collections. EtOH protects DNA by dehydrating and denaturing proteins, thus slowing chemical and enzymatic reactions [15]. While EtOH at 70–95% is inexpensive, easy to obtain, and effective for short-term DNA preservation in biological materials, it is less effective for long-term tissue storage [16]. Additionally, EtOH is flammable and subject to shipping restrictions [8,17].

We chose to evaluate EDTA as it is a common ingredient in many preservative solutions [8] and is the primary active ingredient [18] in DESS [19], a widely used tissue preservative that is as or more effective than EtOH in preventing DNA degradation in tissues from a wide range of organisms [9,18–27]. EDTA reduces DNA degradation in tissues by chelating divalent cations that are required as cofactors by deoxyribonucleases (DNases) [17,18,28,29]. Although EDTA is used most frequently at pH 7.5–8, e.g., in DESS [19], a recent investigation found that EDTA solutions become significantly more effective at preserving HMW DNA when the pH of the solution is raised from 8 to 10, especially for tissues that display rapid DNA degradation in other preservatives [17]. This increased effectiveness correlates with EDTA's increasing capacity to chelate divalent cations as pH rises [17]. EDTA is inexpensive, readily available, and has low toxicity, but unlike EtOH, it is nonflammable and can be shipped as a non-hazardous material [17].

In this study, we evaluate whether treatment with EDTA or EtOH can improve the recovery of HMW DNA from frozen tissues. Specifically, we thawed the previously frozen tissues of 10 marine species (five fishes and five invertebrates) in the preservatives EDTA (250 mM, pH 10) or EtOH (95%) and maintained them in their respective preservatives overnight at 4°C before DNA extraction. We then determined the percentage of high molecular weight DNA (%HMW) and yield of HMW DNA normalized by tissue weight (nY) in each DNA extract and compared these among treatments.

## Materials and methods

### Ethics statement

This study followed the guidelines recommended by the National Institutes of Health Guide for the Care and Use of Laboratory Animals and Northeastern University's Institutional Animal Care and Use Committee policies. Although no live vertebrate animals were used in this study, the general principles of humane animal care were applied to the treatment of live invertebrate animals.

### Specimen selection, storage, and sampling

In this investigation, we sampled a wide range of species, storage conditions, and tissues to reflect a diverse selection of materials from which researchers may wish to isolate high molecular weight DNA.

We chose 16 species that represent a wide range of animal taxa, including members of 16 genera, 12 families, and three phyla (S1 Table). DNA and images of all specimens have been deposited into the Ocean Genome Legacy Center Genomic Resource Collection [30] and are available via the Arctos database (arctosdb.org) [31] under the catalog numbers listed in S1 Table.

To reflect the diversity of cryostorage conditions found across many frozen collections, we selected organisms that were frozen whole and stored for various periods and under various conditions, some of which are unknown. Specifically, 10 specimens each of five marine fish and five marine invertebrate species (S1 Table) were used for the qualitative and quantitative comparison of treatments (Fig 1). Specimens of *Cololabis saira* (Pacific saury), *Larimichthys polyactis* (redlip croaker), *Odontesthes regia* (silverside), *Sardina pilchardus* (European pilchard), *Amphioctopus* sp. (octopus), *Magallana gigas* (Pacific oyster), and *Penaeus vannamei*

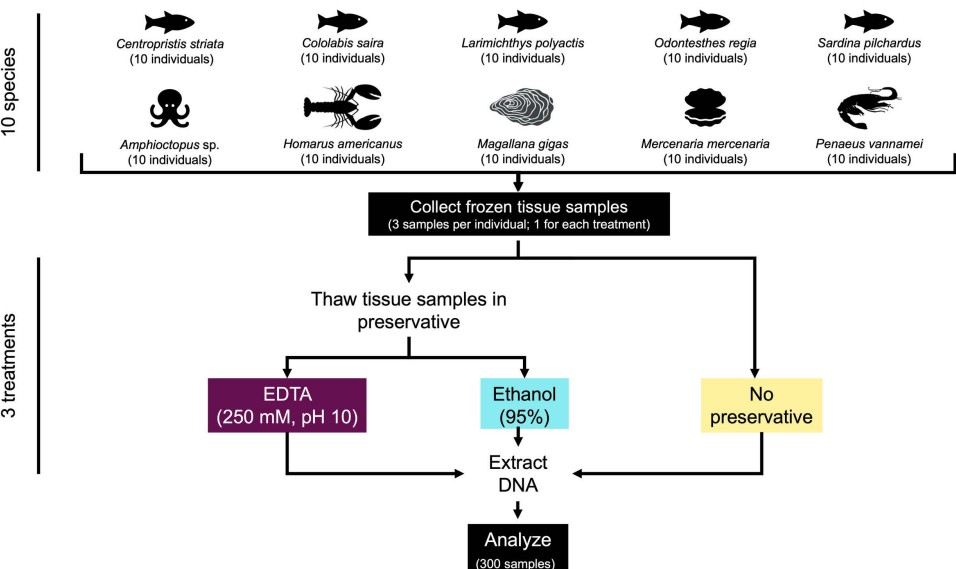

**Fig 1. Experimental Design.** Three frozen tissue samples were collected from each of 10 individuals of 10 marine species (five marine fishes and five marine invertebrates) for a total of 300 samples. One sample from each individual was then thawed in EDTA (250 mM, pH 10) overnight at 4°C and the second was thawed in ethanol (95%) overnight at 4°C, after which DNA was extracted, analyzed, and compared to DNA extracted directly from the third frozen tissue sample, which did not receive liquid preservative treatment.

(whiteleg shrimp) were purchased as whole, frozen individuals at grocery stores and stored at -80°C for 0–43 days before sampling. Specimens of *Centropristis striata* (black sea bass) were collected in trawls by the Massachusetts Department of Marine Fisheries and maintained as frozen whole specimens at -23°C for 9.8 to 9.9 years before transfer to -80°C for 1 day before tissue sampling. Specimens of *Homarus americanus* (American lobster) and *Mercenaria mercenaria* (hard-shell clam) were obtained live from seafood suppliers and frozen at -80°C for 2.1 years before sampling. Specimens of *H. americanus* and *M. mercenaria* were euthanized by splitting, i.e., cutting quickly along the longitudinal midline with a large sharp knife. While proposed euthanasia methods are published for some invertebrate taxa, there is a lack of peer-reviewed literature evaluating their effectiveness [32]. Nonetheless, the methods described here have been judged as likely to be humane for lobsters [33]. To our knowledge, reliable data on euthanasia are not available for the bivalve *M. mercenaria.*

Specimens of six additional fish species, *Alosa mediocris* (hickory shad; *N* = 3), *Brevoortia tyrannus* (Atlantic menhaden; *N* = 6), *Cynoscion regalis* (weakfish; *N* = 3), *Peprilus triacanthus* (Atlantic butterfish; *N* = 3), *Scomberomorus maculatus* (Spanish mackerel; *N* = 6), and *Trinectes maculatus* (hogchoker; *N* = 4), were collected in trawls by the Connecticut Department of Energy and Environmental Protection and were maintained as frozen whole specimens for 2.8 to 3.9 years at -80°C before tissue sampling. Because too few specimens were available to achieve statistical power sufficient for preservative treatment comparisons, specimens of these species were used only for qualitative analyses.

The samples we collected also reflected a range of tissues. Samples from all fish specimens were collected from the dorsal musculature posterior to the pectoral fin. Tissue samples from *Amphioctopus* sp., *Magallana gigas*, and *Mercenaria mercenaria* were collected from tentacles, visceral mass, and mantle, respectively. For *H. americanus* and *Penaeus vannamei*, tissue samples were collected from the abdominal musculature.

## Tissue handling and DNA extraction

All specimens were dissected while frozen on a chilled aluminum plate to minimize thawing. For specimens of all species, excluding *H. americanus* and *M. mercenaria*, three tissue samples of 100 mg (avg. = 102.0 ± 7.9 mg) each were rapidly collected from each frozen specimen, placed into 1.5 mL microcentrifuge tubes, and immediately stored at -80°C for 5–31 days before subsequent processing. For each of these specimens, 1 mL of EDTA (250 mM, pH 10) was added to the first of the three frozen tissue samples and 1 mL of EtOH (95%) was added to the second. These samples were stored in their respective preservatives overnight at 4°C. Subsequently, a 25 mg (avg. = 25.5 ± 3.4 mg) subsample was removed from each 100 mg sample and transferred to the Qiagen DNeasy Blood and Tissue kit (Hilden, Germany) lysis solution for DNA extraction. The third tissue sample from each frozen specimen was dissected on a chilled aluminum plate and a 25 mg (avg. = 25.9 ± 4.1 mg) subsample was transferred directly into the lysis solution without additional liquid preservative treatment. For *H. americanus* and *M. mercenaria*, 100 mg (avg. = 99.9 ± 16.4 mg) samples were handled as above except *(1)* tissues were processed immediately rather than being stored at -80°C before 25 mg (avg. = 25.4 ± 1.7 mg) subsamples were taken for DNA extraction and *(2)* frozen 25 mg (avg. = 25.4 ± 1.4 mg) tissue subsamples with no subsequent liquid preservative treatment were taken directly from frozen specimens for DNA extractions. Samples for all species, excluding *Centropristis striata, H. americanus,* and *M. mercenaria*, were re-weighed after preservative treatment but before subsampling. We used this post-treatment weight to calculate changes in mass due to preservative treatment.

DNA was extracted using the Qiagen DNeasy Blood and Tissue kit (Hilden, Germany) according to the manufacturer's protocol, with overnight digestions at 56°C and the following modifications: *(1)* after digestion, 4 µL of RNase A (100 mg/mL; Qiagen; Hilden, Germany) were added to each subsample and subsamples were incubated at room temperature for 2 minutes before proceeding with the extraction protocol and *(2)* DNA was eluted with two sequential 50 µL volumes of buffer AE and combined for a total of 100 µL.

## DNA analysis

For qualitative visualization, 3 µL (between 4.98 and 720 ng DNA) of each purified DNA sample were size-separated by gel electrophoresis (0.8% agarose, 1X TAE, 5 volts/cm for 50 minutes), stained with 1% GelRed nucleic acid stain (Biotium; Fremont, CA), and imaged using a Gel Doc XR+ Molecular Imager (Bio-Rad; Hercules, CA). The first two and last two lanes of each gel were loaded with 0.33, 0.66, or 1 µL of Quick-Load Purple 1 kb Plus DNA Ladder (100 µg/mL; New England Biolabs; Ipswich, MA) as molecular weight standards. The specific volumes of ladder used are indicated in each figure legend.

To determine %HMW and nY, we analyzed 1 µL (between 1.66 and 240 ng DNA) of each DNA extract using the Agilent Technologies TapeStation 2200 DNA Analyzer (Santa Clara, CA) with Genomic DNA ScreenTapes and TapeStation Analysis Software version A.02.02 (SR1). In this investigation DNA fragments greater than 10 kb are considered HMW and fragments from 150 bp to 10 kb are considered low molecular weight (LMW). As in [17], we estimated the %HMW for each sample as 100 minus the percentage of low molecular weight DNA (%LMW) as measured by the DNA Analyzer. This method avoids the exclusion of DNA fragments greater than the 60 kb resolution limit of the DNA Analyzer from the %HMW estimate.

$$\%HMW = \left(100 - \%LMW\right) = 100 - \left(\frac{LMW\ DNA\ concentration\left(ng\ /\ \mu L\right)}{total\ DNA\ concentration\left(ng\ /\ \mu L\right)} * 100\right)$$

The normalized yield of HMW DNA (nY) was calculated for each sample by multiplying the %HMW by the total amount of DNA in the sample (%HMW x concentration x elution volume), then dividing that product by the weight of the tissue used for the extraction multiplied by a correction ratio (CR) to account for the effect of treatment on tissue mass.

$$nY = \frac{\%HMW * total\ DNA\ concentration\left(ng\ /\ \mu L\right) * elution\ volume\left(\mu L\right)}{weight\ of\ tissue\ used\ for\ the\ extraction\left(mg\right) * 1000\left(ng\ /\ mg\right) * CR}$$

For all species, excluding *C. striata*, *H. americanus*, and *M. mercenaria,* CR was determined by dividing the initial weight of the whole sample by its post-treatment weight [18].

$$CR = \frac{initial\ sample\ weight\left(mg\right)}{post\text{-}treatment\ sample\ weight\left(mg\right)}$$

For *C. striata*, *H. americanus*, and *M. mercenaria,* a post-treatment weight was not determined, and therefore, tissue weights were not corrected (i.e., CR = 1).

To evaluate DNA purity, optical absorbance was measured for 2 µL of each sample (between 3.32 and 480 ng DNA) at 230, 260, and 280 nm using a Nanodrop 1000 droplet spectrophotometer (Thermo Fisher Scientific; Waltham, MA) and the $A_{260}/A_{280}$ and $A_{260}/A_{230}$ absorbance ratios were calculated.

All statistics were analyzed and visualized in RStudio [34,35] using the tidyverse version 2.0.0 [36], conover.test version 1.1.6 [37], multcompView version 0.1–10 [38], cowplot version

1.1.3 [39], stringr version 1.5.1 [40], and rstatix version 0.7.2 [41] packages. The effect of preservative treatment on %HMW and nY values were analyzed independently for each species. To determine whether the data followed a normal distribution, a Shapiro-Wilk normality test was performed. If the data were normally distributed, Mauchly's test of sphericity was performed, followed by a one-way repeated measures ANOVA and a Bonferroni-corrected Tukey post-hoc test. If the data were not normally distributed, a Friedman $\chi^2$ test was performed, followed by a Bonferroni-corrected Conover post-hoc test.

### PCR amplification and DNA sequencing

The suitability of the extracted DNA for typical downstream applications was evaluated as in [17]. Briefly, the barcode segment [42] of the cytochrome c oxidase subunit 1 (*COI*) gene from each DNA extract was amplified by polymerase chain reaction (PCR) and Sanger sequenced. PCR amplifications for all fish species used the primers FISHCOILBC (5'–TCAACYAATCAYAAAGATATYGGCAC–3') and FISHCOIHBC (5'–ACTTCYGGGT-GRCCRAARAATCA–3') [43]. For invertebrate species, primers LCO1490_t1 (5'–TGTA-AAACGACGGCCAGTGGTCAACAAATCATAAAGATATTGG–3') and HCO2198_t2 (5'–CAGGAAACAGCTATGACTAAACTTCAGGGTGACCAAAAAATCA–3') [44] were used. Each reaction consisted of 17.5 µL of One*Taq* 2X Master Mix with Standard Buffer (New England Biolabs; Ipswich, MA), 14.1 µL of ultrapure water, 2 µL of DNA template, and 0.7 µL of each of the forward and reverse primers. The thermocycling conditions for fish species were: 30 seconds at 94°C; 30 cycles of 30 seconds at 94°C, 40 seconds at 52°C, and 60 seconds at 68°C; 5 minutes at 68°C; hold at 4°C. The thermocycling conditions for invertebrate species were: 30 seconds at 94°C; 34 cycles of 30 seconds at 94°C, 50 seconds at 45°C, and 60 seconds at 68°C; 5 minutes at 68°C; hold at 4°C. PCR success was evaluated by gel electrophoresis, as described above; when appropriately sized bands were visualized on the gel, the PCR amplification was considered successful.

All PCR products were bidirectionally sequenced by Sanger sequencing at Psomagen (Rockville, MD) using an ABI 3730XL Analyzer (Applied Biosystems; Foster City, CA). Sequences were analyzed with Geneious Prime (Auckland, New Zealand). Ends were automatically trimmed to remove primers sequences and sequence regions with greater than 1% error probability. Sequences greater than 500 bp in length and with a greater than 98% Q20 score were considered successful. Assembled contigs and sequences were deposited into the GenBank [45] database under the accession numbers listed in S1 Table.

## Results

DNA was recovered from all samples, species, and treatments; %HMW values ranged from 3.85 to 86.21% and nY values ranged from 0.00030 to 0.44367 µg DNA/mg tissue (S2 Table). Treatment significantly affected %HMW and nY in eight and six of the 10 statistical models, respectively (Figs 2 and 3; Table 1). The absorbance and absorbance ratios $A_{260}$, $A_{260}/A_{280}$, and $A_{260}/A_{230}$ and Tapestation DNA Analyzer data are presented in S2 Table.

### Effect of treatments on %HMW

DNA extracts from muscle tissue samples of all five of the fish species examined quantitatively (*Centropristis striata, Cololabis saira, L. polyactis, O. regia,* and *Sardina pilchardus*) yielded significantly greater %HMW values when thawed in EDTA before DNA extraction as compared to those isolated directly from frozen tissues (Fig 2). This effect was most extreme in tissues of *Centropristis striata,* where DNA extracts from EDTA-thawed tissues yielded an average %HMW value more than three-fold higher than extracts from frozen tissues (54.8 ±

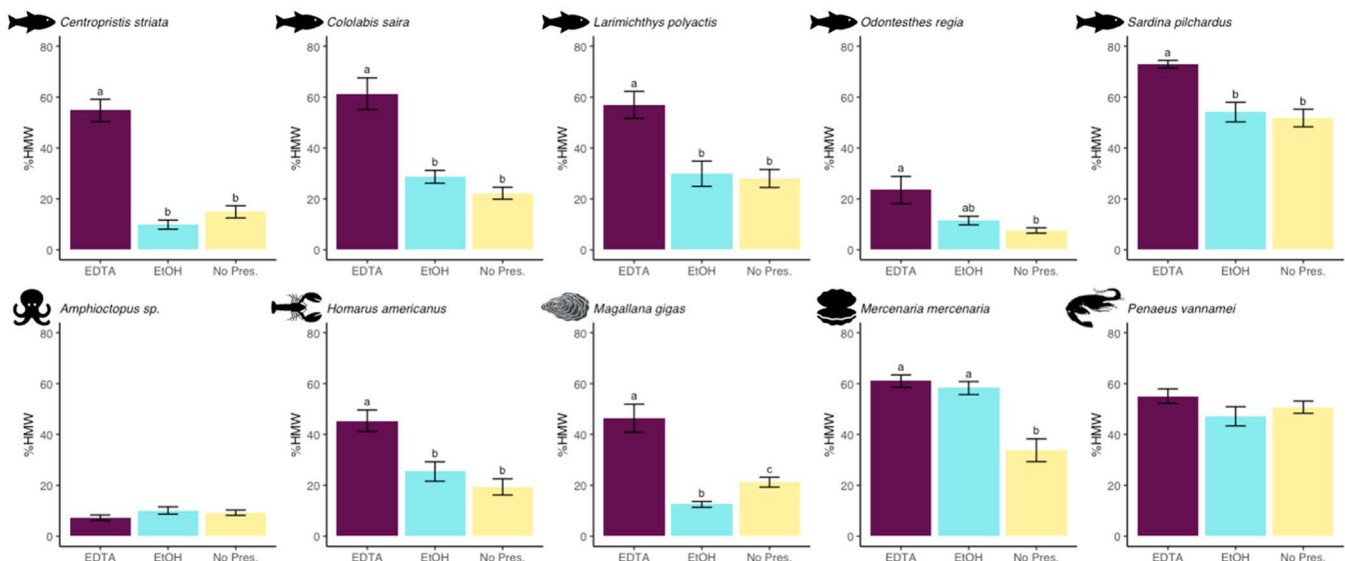

**Fig 2. Percentage of high molecular weight DNA.** Average values for percentage of high molecular weight DNA (%HMW) were determined for extracts prepared from frozen tissue samples collected from each of 10 individuals of 10 marine species (five marine fishes and five marine invertebrates). One sample from each individual was then thawed in EDTA (250 mM, pH 10; maroon) overnight at 4°C and the second was thawed in ethanol (95%; blue) overnight at 4°C, after which DNA was extracted, analyzed, and compared to DNA extracted directly from the third frozen tissue sample, which did not receive liquid preservative treatment (yellow). Error bars represent standard error. Within each histogram, treatments bearing different lower-case letters are significantly different at $p < 0.05$; matching lower-case letters indicate statistically indistinguishable treatments; an absence of letters indicates no significant differences among all treatments in a given model.

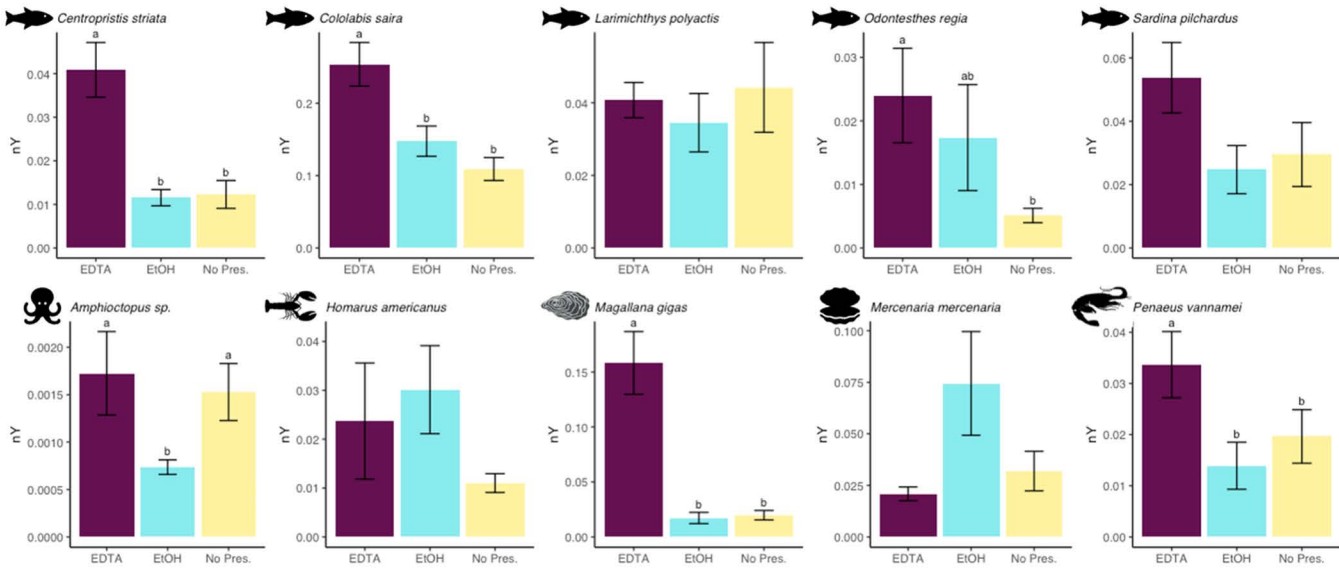

**Fig 3. Normalized yield of high molecular weight DNA.** Average values for yields of high molecular weight DNA normalized by tissue weight (nY) were determined for extracts prepared from frozen tissue samples collected from each of 10 individuals of 10 marine species (five marine fishes and five marine invertebrates). One sample from each individual was then thawed in EDTA (250 mM, pH 10; maroon) overnight at 4°C and the second was thawed in ethanol (95%; blue) overnight at 4°C, after which DNA was extracted, analyzed, and compared to DNA extracted directly from the third frozen tissue sample, which did not receive liquid preservative treatment (yellow). Error bars represent standard error. Within each histogram, treatments bearing different lower-case letters are significantly different at $p < 0.05$; matching lower-case letters indicate statistically indistinguishable treatments; an absence of letters indicates no significant differences among all treatments in a given model. Note that y-axis scales differ among species.

**Table 1. The effect of preservative treatments on average percentages of high molecular weight DNA (%HMW) and average yields of high molecular weight DNA normalized by tissue weight (nY) for DNA extracts from frozen tissues of 10 specimens of each of 10 marine species.**

| | Species | Statistical Test | Test Statistic | df | *p*-value |
|---|---|---|---|---|---|
| **%HMW** | *Centropristis striata* | Friedman $\chi^2$ | $\chi^2 = 16.8$ | 2 | < 0.001* |
| | *Cololabis saira* | Friedman $\chi^2$ | $\chi^2 = 15.8$ | 2 | < 0.001* |
| | *Larimichthys polyactis* | Repeated measures ANOVA | $F = 22.541$ | 2 | < 0.001* |
| | *Odontesthes regia* | Friedman $\chi^2$ | $\chi^2 = 10.4$ | 2 | 0.006* |
| | *Sardina pilchardus* | Repeated measures ANOVA | $F = 15.295$ | 2 | < 0.001* |
| | *Amphioctopus* sp. | Friedman $\chi^2$ | $\chi^2 = 2.6$ | 2 | 0.273 |
| | *Homarus americanus* | Repeated measures ANOVA | $F = 26.577$ | 2 | < 0.001* |
| | *Magallana gigas* | Friedman $\chi^2$ | $\chi^2 = 16.8$ | 2 | < 0.001* |
| | *Mercenaria mercenaria* | Repeated measures ANOVA | $F = 53.563$ | 2 | < 0.001* |
| | *Penaeus vannamei* | Repeated measures ANOVA | $F = 1.376$ | 2 | 0.278 |
| **nY** | *Centropristis striata* | Friedman $\chi^2$ | $\chi^2 = 15$ | 2 | < 0.001* |
| | *Cololabis saira* | Friedman $\chi^2$ | $\chi^2 = 7.2$ | 2 | 0.027* |
| | *Larimichthys polyactis* | Friedman $\chi^2$ | $\chi^2 = 2.4$ | 2 | 0.301 |
| | *Odontesthes regia* | Friedman $\chi^2$ | $\chi^2 = 9.8$ | 2 | 0.007* |
| | *Sardina pilchardus* | Friedman $\chi^2$ | $\chi^2 = 1.4$ | 2 | 0.497 |
| | *Amphioctopus* sp. | Friedman $\chi^2$ | $\chi^2 = 9.6$ | 2 | 0.008* |
| | *Homarus americanus* | Friedman $\chi^2$ | $\chi^2 = 3.8$ | 2 | 0.150 |
| | *Magallana gigas* | Friedman $\chi^2$ | $\chi^2 = 12.2$ | 2 | 0.002* |
| | *Mercenaria mercenaria* | Friedman $\chi^2$ | $\chi^2 = 3.8$ | 2 | 0.150 |
| | *Penaeus vannamei* | Friedman $\chi^2$ | $\chi^2 = 12.2$ | 2 | 0.002* |

Significant *p*-values ($p < 0.05$) indicated with

*; df, degrees of freedom.

13.9% vs. 14.9 ± 7.6%). DNA extracts from EDTA-thawed tissues also yielded higher %HMW values than those from EtOH-thawed tissues for four of the five fish species (*Centropristis striata, Cololabis saira, L. polyactis,* and *S. pilchardus*). No significant differences in %HMW values were observed between DNA extracts obtained from EtOH-thawed tissues and those obtained directly from frozen tissues among these five fish species.

Similarly, DNA extracts from tissues of three of the five invertebrate species (*H. americanus, Magallana gigas,* and *Mercenaria mercenaria*) yielded significantly greater %HMW values when isolated from tissues thawed in EDTA compared to those purified directly from frozen tissues (Fig 2). This difference was greatest in *H. americanus*, where tissues thawed in EDTA yielded an average %HMW value more than two-fold greater than extracts prepared directly from frozen tissues (45.3 ± 13.3% vs. 19.4 ± 10.0%). DNA extracts from tissues of two of the five invertebrate species (*H. americanus* and *Magallana gigas*) showed significantly greater %HMW values when thawed in EDTA than when thawed in EtOH. For four of the five invertebrate species (*Amphioctopus* sp.*, H. americanus, M. gigas,* and *P. vannamei*), %HMW values of DNA extracts from EtOH-thawed tissues were not significantly different or significantly lower than those from frozen tissues.

For all 10 species, we observed no significant decreases in %HMW values in extracts of EDTA-thawed tissues compared to those extracted directly from frozen tissues.

### Effect of treatments on nY

Patterns observed for nY were similar to those seen in %HMW. For three of the five fish species (*Centropristis striata, Cololabis saira,* and *O. regia*), the nY values of DNA extracts from muscle tissue samples were significantly greater when thawed in EDTA before extraction as compared to those isolated directly from frozen tissues (Fig 3). This effect was most extreme in tissues of *O. regia*, where the average nY value of DNA extracts from EDTA-thawed tissues was more than four-fold higher than extracts from frozen tissues (0.02400 ± 0.02350 μg DNA/ mg tissue vs. 0.00511 ± 0.00358 μg DNA/mg tissue). The nY values of DNA extracts from EDTA-thawed tissues were also greater than those from EtOH-thawed tissues for two of the five fish species (*Centropristis striata* and *Cololabis saira*). No significant differences in nY values were observed between DNA extracts obtained from EtOH-thawed tissues and those obtained directly from frozen tissues for all five of the fish species.

Similarly, for two of the five invertebrate species (*M. gigas* and *P. vannamei*), the nY values of DNA extracts were significantly greater when isolated from tissues thawed in EDTA compared to those purified directly from frozen tissues (Fig 3). This difference was greatest in *P. vannamei*, where the average nY value of tissues thawed in EDTA was nearly eight-fold greater than extracts prepared directly from frozen tissues (0.15825 ± 0.09017 μg DNA/mg tissue vs. 0.01981 ± 0.01359 μg DNA/mg tissue). The nY values of DNA extracts from tissues of three of the five invertebrate species (*Amphioctopus* sp.*, M. gigas,* and *P. vannamei*) were significantly greater when thawed in EDTA than when thawed in EtOH. The nY values of DNA extracted from frozen tissues were significantly greater than those from tissues thawed in EtOH for only one species (*Amphioctopus* sp.).

For all 10 species, we observed no significant decreases in nY values in extracts of EDTA-thawed tissues compared to those extracted directly from frozen tissues.

### Qualitative DNA analyses

The quantitative results for the 10 species described above are consistent with qualitative observations made by gel electrophoresis (S1 Fig). Gel electrophoresis also revealed qualitatively similar trends for DNA extracts from tissues of six additional fish species for which too few specimens were available for conclusive statistical analyses (S2 Fig).

### PCR amplification and DNA sequencing

To determine if DNA extracts were of sufficient quality for downstream applications, PCR amplification and *COI* sequencing success were evaluated for all 16 species. Across all species and treatments, 96% of PCR amplifications were successful as determined by the presence of a single appropriately sized band on an electrophoretic gel (S2 Table). Specifically, only 1 of 125 (0.8%) reactions from extracts of EDTA-thawed tissues was unsuccessful, whereas 6 (4.8%) and 8 (6.4%) reactions from extracts of EtOH-thawed and frozen tissues were unsuccessful, respectively. These failed reactions were limited to three species: *Centropristis striata*, *T. maculatus*, and *Amphioctopus* sp.

The overall success rate of sequencing was also high, with 92.5% of reactions producing sequences longer than 500 bp and with Q20 scores greater than 98% (S2 Table). Specifically, 7 (5.6%), 11 (8.8%), and 10 (8.0%) of 125 reactions each from extracts of EDTA-thawed, EtOH-thawed, and frozen tissues were unsuccessful, respectively. These failed reactions were limited to eight species—six fishes and two invertebrates. Note that all sequencing reactions for *T.*

*maculatus* were unsuccessful, regardless of treatment. In all cases, successful sequences from the same organism were identical regardless of treatment. Additionally, sequencing revealed that two species of octopus, *Amphioctopus aegina* (*N* = 8) and *A. fangsiao* (*N* = 2), were present among our specimens although they were all purchased as *A. membranaceus*. These are closely related sister species and so were grouped together as *Amphioctopus* sp. in our analysis.

## Discussion

Cryopreservation is one of the most common ways that individual researchers and curators of biological collections preserve tissues for genetic and genomic research applications [2]. Under optimal conditions, cryopreservation can indefinitely maintain DNA integrity in tissues [8,46,47]. Storage in liquid nitrogen dewars (-196°C) is considered best for HMW DNA preservation, effectively halting enzymatic and chemical activity [5]. Storage in ultracold freezers (-80°C) dramatically reduces but does not entirely halt enzymatic activity, so gradual DNA degradation may still occur [8,46,48]. Nonetheless, storage at -80°C is an excellent option that can preserve DNA in tissues for decades [48]. Storage at -20°C in standard laboratory freezers may also be suitable for shorter-term preservation of HMW DNA in tissues [8].

However, because most DNA extractions are performed in liquid media, frozen tissue samples must typically be thawed either before or during DNA extraction, potentially exposing their DNA to chemical or enzymatic degradation. In this investigation, we show that thawing frozen tissues in EDTA (250 mM, pH 10) and maintaining them in this preservative overnight at 4°C can significantly improve the recovery of HMW DNA from those tissues. For eight of the 10 fish and invertebrate species examined, %HMW values were up to three-fold greater in DNA extracts from EDTA-treated tissues compared to extracts from untreated frozen tissues. Similarly, nY values were improved up to eight-fold in extracts from five of the 10 species examined compared to extracts prepared directly from frozen tissues. Indeed, we observed significant improvement in %HMW, nY, or both in nine of the 10 species examined.

Importantly, we observed no significant decreases in %HMW or nY values for extracts of EDTA-thawed tissues compared to those extracted directly from frozen tissues, nor were harmful effects observed on PCR amplification or DNA sequencing. Indeed, success rates for PCR amplification and Sanger sequencing of the mitochondrial *COI* gene were highest for DNA extracts from tissues thawed in EDTA.

Furthermore, EDTA treatment was effective in improving recovery of HMW DNA from a variety of frozen tissue types across a broad range of taxa. Significant improvements in %HMW, nY, or both were observed in tissue types derived from dorsal musculature, tentacles, mantle, and abdominal musculature from representatives of three phyla (Chordata, Mollusca, and Arthropoda), including 5 fish and 4 invertebrate species. Additionally, we saw qualitative evidence for reduced DNA degradation in extracts of EDTA-thawed samples from an additional six fish species where statistical analyses were not performed.

In contrast, EtOH treatment provided little or no protection for DNA during the transition from frozen tissue to purified DNA. Specifically, for nine of the 10 species examined, average %HMW values for extracts from EtOH-thawed tissues were not significantly different or were significantly lower than those from frozen tissues. The same is true for nY in all 10 species.

Interestingly, thawing frozen tissue samples in EDTA prior to DNA extraction improved recovery of HMW DNA from frozen tissues maintained under a range of cryostorage conditions. For example, %HMW and nY values were significantly improved for *H. americanus* and *Mercenaria mercenaria*, which were stored at -80°C for two years, as well as *C. striata*, which was stored under less ideal conditions at -23°C for nine years. These results indicate that

regardless of freezer storage conditions, EDTA treatment can improve the recovery of HMW DNA.

In conclusion, our results indicate that, although cryopreservation can provide excellent protection for HMW DNA, substantial DNA degradation can occur during the preparation of frozen tissues for DNA extraction. In these experiments, we took care to minimize the thawing of frozen tissues before DNA extraction by working quickly and subsampling tissues on a chilled aluminum plate. Nonetheless, we observed substantial DNA degradation in most extracts prepared directly from frozen tissues. While this degradation might have been reduced by performing the tissue dissections under colder conditions, e.g., in liquid nitrogen or on dry ice, such stringent conditions are difficult, inconvenient, dangerous, and expensive to maintain. Our results show that thawing frozen tissues in EDTA provides a convenient alternative that reduces DNA degradation and allows tissues to be safely subsampled, weighed, and handled at room temperature with significantly reduced DNA degradation.

Finally, we note that tissues of different types and taxonomic origin may vary widely in physical, chemical, and biological properties, e.g., hardness, surface area to volume ratio, permeability, pH, chemical composition, enzymatic activity, etc., and that these properties can influence the performance of preservative treatments. Nonetheless, in this investigation, EDTA improved the recovery of HMW DNA across a wide variety of marine animal species and tissues, suggesting the potential of this method for broader application. Further investigation is warranted to determine if EDTA treatment can improve DNA recovery from frozen tissues of other marine and non-marine organisms across the tree of life, thereby adding value to the enormous variety of samples held in frozen research collections worldwide.

## Supporting information

**S1 Fig. Visualization of DNA extracts by gel electrophoresis for 10 marine species.** DNA was extracted from frozen tissue samples collected from each of 10 individuals of 10 marine species (five marine fishes and five marine invertebrates) that were thawed in EDTA (250 mM, pH 10; lanes 1–10) or ethanol (95%; lanes 11–20) overnight at 4°C or extracted directly from frozen tissues without subsequent liquid preservative treatment (lanes 21–30). Lanes marked with an L contain 0.66 µL of Quick Load Purple 1 kb Plus DNA Ladder (100 µg/mL; New England Biolabs; Ipswich, MA), except for *Centropristis striata*, for which lanes marked with an L contain 1 µL of Quick Load Purple 1 kb Plus DNA Ladder. Specimens are presented in the same order across all treatments.
(PDF)

**S2 Fig. Qualitative visualization of DNA extracts from six additional fish species by gel electrophoresis.** DNA was extracted from tissues of six additional marine fish species that were thawed in EDTA (250 mM, pH 10; lanes 1–2) or ethanol (95%; lanes 3–4) overnight at 4°C or extracted directly from frozen tissues without subsequent liquid preservative treatment (lanes 5–6) from two randomly selected specimens of each species. Lanes marked with an L contain 0.66 µL of Quick Load Purple 1 kb Plus DNA Ladder (100 µg/mL; New England Biolabs; Ipswich, MA). Specimens are presented in the same order across all treatments.
(PDF)

**S3 Fig. Raw gel images for S1 Fig.** DNA was extracted from frozen tissue samples collected from each of 10 individuals of 10 marine species (five marine fishes and five marine invertebrates) that were thawed in EDTA (250 mM, pH 10; lanes 21–30) or ethanol (95%; lanes 11–20) overnight at 4°C or extracted directly from frozen tissues without subsequent liquid preservative treatment (lanes 1–10). Lanes marked with an L contain either 0.33 or 0.66 µL of

Quick Load Purple 1 kb Plus DNA Ladder (100 μg/mL; New England Biolabs; Ipswich, MA), except for *Centropristis striata*, for which lanes marked with an L contain 1 μL of Quick Load Purple 1 kb Plus DNA Ladder. Specimens are presented in the same order across all treatments. Dashes indicate empty lanes.
(PDF)

**S4 Fig. Raw gel images for S2 Fig.** DNA was extracted from tissues of six additional marine fish species that were thawed in EDTA (250 mM, pH 10; lanes 3–4) or ethanol (95%; lanes 5–6) overnight at 4°C or extracted directly from frozen tissues without subsequent liquid preservative treatment (lanes 1–2) from two randomly selected specimens of each species. Lanes marked with an L contain 0.66 μL of Quick Load Purple 1 kb Plus DNA Ladder (100 μg/ mL; New England Biolabs; Ipswich, MA). Specimens are presented in the same order across all treatments.
(PDF)

**S1 Table. Specimen information.** Taxonomic, source, collection, and storage information as well as Ocean Genome Legacy (OGL) catalog and NCBI accession numbers are presented for specimens of 16 marine fish and invertebrate species. N/a indicates samples for which NCBI accession numbers were not assigned due to sequencing failures.
(PDF)

**S2 Table. Supporting data.** Supporting data are presented for DNA extracts of tissues from specimens of 16 marine fish and invertebrate species that were thawed in EDTA (250 mM, pH 10) or ethanol (95%) overnight at 4°C or extracted directly from frozen tissues without subsequent liquid preservative treatment. Specifically, species, sample ID, specimen and replicate numbers, treatments, initial and post-treatment sample weights, correction ratios, measured and corrected weights of tissue subsamples used for DNA extraction, $A_{260}$, $A_{260}/A_{280}$ ratios, and $A_{260}/A_{230}$ ratios calculated using the Nanodrop 1000 droplet spectrophotometer, total and low molecular weight (150 bp–10kb) DNA concentration values determined by the Agilent Tapestation DNA Analyzer 2200, percentages of low molecular weight (%LMW) and high molecular weight (%HMW) DNA and nY calculated based on Tapestation data, as well as mitochondrial *COI* PCR amplification and sequencing success are presented for each DNA sample analyzed in this study. N/a indicates samples for which data were not collected.
(PDF)

## Acknowledgments

We thank Bob and Eileen Matz, the Comb family, Emily Condon and the Northeastern University co-op program, the Northeastern University Marine Science Center community, and the many others who have supported the Ocean Genome Legacy Center at Northeastern University. We would also like to thank the Connecticut Department of Energy and Environmental Protection, the National Oceanic and Atmospheric Administration Fisheries, and the Grabowski lab at Northeastern University for donating specimens used in this study.

## Author contributions

**Conceptualization:** Ella Messner, Lev Becker, Mia L. DeSanctis, Elizabeth A. Soranno, Rosalia Falco Poulin, Hannah J. Appiah-Madson, Daniel L. Distel.

**Data curation:** Ella Messner, Lev Becker, Rosalia Falco Poulin, Hannah J. Appiah-Madson.

**Formal analysis:** Ella Messner, Lev Becker, Rosalia Falco Poulin, Hannah J. Appiah-Madson.

**Funding acquisition:** Daniel L. Distel.

**Investigation:** Ella Messner, Lev Becker, Mia L. DeSanctis, Elizabeth A. Soranno, Ryan Pianka, Caileigh Pierce, Molly Johnson.

**Methodology:** Ella Messner, Lev Becker, Mia L. DeSanctis, Elizabeth A. Soranno, Rosalia Falco Poulin, Hannah J. Appiah-Madson, Daniel L. Distel.

**Project administration:** Rosalia Falco Poulin, Hannah J. Appiah-Madson, Daniel L. Distel.

**Supervision:** Rosalia Falco Poulin, Hannah J. Appiah-Madson, Daniel L. Distel.

**Visualization:** Lev Becker, Rosalia Falco Poulin, Hannah J. Appiah-Madson, Daniel L. Distel.

**Writing – original draft:** Ella Messner, Lev Becker, Rosalia Falco Poulin, Hannah J. Appiah-Madson, Daniel L. Distel.

**Writing – review & editing:** Ella Messner, Lev Becker, Mia L. DeSanctis, Ryan Pianka, Molly Johnson, Rosalia Falco Poulin, Hannah J. Appiah-Madson, Daniel L. Distel.

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
