## [Decision Letter · Decision Letter 0]

8 Jan 2025

PONE-D-24-43598Perish the thawed? EDTA reduces DNA degradation during extraction from frozen tissue.PLOS ONE

Dear Dr. Distel,

Thank you for submitting your manuscript to PLOS ONE. After careful consideration, we feel that it has merit but does not fully meet PLOS ONE’s publication criteria as it currently stands. Therefore, we invite you to submit a revised version of the manuscript that addresses the points raised during the review process.

We look forward to receiving your revised manuscript.

Kind regards,

Shailender Kumar Verma, Ph.D.

Academic Editor

PLOS ONE

**Journal Requirements:**

2. To comply with PLOS ONE submissions requirements, in your Methods section, please provide additional information regarding the experiments involving animals and ensure you have included details on (a) methods of sacrifice, (b) methods of anesthesia and/or analgesia, and (c) efforts to alleviate suffering.

Reviewers' comments:

Reviewer's Responses to Questions

**Comments to the Author**

1. Is the manuscript technically sound, and do the data support the conclusions?

Reviewer #1: Yes

Reviewer #2: Partly

2. Has the statistical analysis been performed appropriately and rigorously? 

Reviewer #1: Yes

Reviewer #2: Yes

3. Have the authors made all data underlying the findings in their manuscript fully available?

Reviewer #1: Yes

Reviewer #2: No

4. Is the manuscript presented in an intelligible fashion and written in standard English?

Reviewer #1: Yes

Reviewer #2: Yes

5. Review Comments to the Author

**Reviewer #1: ** The manuscript addresses an important issue in DNA preservation during tissue thawing, offering a potential solution using EDTA. The study's relevance is significant for genomic and genetic research, especially in cryopreserved biological samples.

There are several questions.

1. Why was the focus placed specifically on EDTA and ethanol (EtOH)? Were other preservatives considered but excluded, and if so, why?

2. What novel insights does this study add beyond previous research on EDTA in DNA preservation?

3�Explain why authors chose these 10 organisms as target species.

4.Were any controls implemented to account for potential variability in tissue type (muscle vs. visceral mass)? Could tissue composition impact the efficacy of EDTA treatment?

5. Is there variability in tissue degradation across species that could confound the interpretation of EDTA's effectiveness? Would standardized storage conditions for all species improve the reliability of cross-species comparisons?

6. What mechanisms might explain the species-specific variations observed in DNA preservation? Expand on potential biological or chemical factors.

7. How does this study inform protocols for non-marine or plant tissues? Discuss the generalizability of findings to other types of biological samples.

**Reviewer #2:**  This study presents a comparative analysis of the effects of EDTA and ethanol on the preservation of high-molecular-weight (HMW) DNA in frozen tissues. The authors hypothesize that EDTA, known for its metal chelation properties, can reduce DNA degradation during the thawing process, which is a common step in DNA extraction that can lead to DNA damage. The study involved ten marine species, with tissues treated with EDTA at pH 10 or ethanol, and then analyzed for HMW DNA recovery. The results indicate that EDTA, particularly at pH 10, significantly improves the recovery of HMW DNA compared with that of ethanol and untreated tissues. These findings suggest that EDTA treatment can be a simple and effective method to increase DNA recovery from frozen tissues, which has implications for genetic and genomic research. Before this manuscript can be published, several questions need to be addressed:

1. Since EDTA is one of the primary active ingredients of DESS, why not compare it with DESS instead of choosing ethanol for comparison?

2. In the Materials and methods section, the authors describe taking 100 mg of tissue and adding EDTA or ethanol, then placing it at 4°C for 12-24 hours before dividing it into 25 mg pieces for DNA extraction. Therefore, what is the purpose of the remaining three 25 mg tissues? If only 25 mg is used for testing, would not it be better to directly immerse the 25 mg tissue into EDTA or ethanol for better protection of DNA? Please provide a detailed explanation.

3. Regarding the above question, does the N=10 in Figure 1 refer to 10 marine fish or 10 marine invertebrate species, or is it 10 samples taken for testing? The S1 Table shows 10 fish, whereas the S2 Table shows 10 measurements for each treatment.

4. Lines 198-200: The authors calculated the concentration of HMW DNA for each sample by subtracting the low molecular weight DNA concentration from the total concentration of DNA present. However, quantitative results of the total DNA concentration and the low molecular weight DNA concentration are lacking. Please supplement more detailed data on how to calculate the concentration of HMW DNA.

5. Lines 195-215 & S2 Table: Table S2 shows only the raw results of the Nanodrop 1000. The %HMW and nY values were calculated from the DNA concentration analyzed via the Agilent Technologies TapeStation 2200 DNA Analyzer. Please provide raw results from Agilent and the detailed data on how to calculate the concentration of HMW DNA. In addition, why are not Agilent's results provided in S2 Fig?

6. Lines 301--304: For four of the five invertebrate species (Amphioctopus sp., H. americanus, M. gigas, and P. vannamei), the %HMW values of DNA extracts from EtOH-thawed tissues were the same as or significantly lower than those from frozen tissues. However, for H. americanus, as shown in Fig. 2, the %HMW value from EtOH-thawed tissues was obviously higher than that from frozen tissues. Please clarify further.

7. Line 159: The samples were stored in preservative for 12 to 24 hours at 4°C. This means that the thawing times of the samples in the preservatives were inconsistent. Therefore, are the data in manuscript accurately compared under the same variable?

8. Lines 147--151: Tissue samples were collected from different tissue parts of different species, such as the musculature posterior to the pectoral fin, tentacles, visceral mass, and mantle. Does the difference between tissue samples affect the determined results and conclusions?

6. PLOS authors have the option to publish the peer review history of their article (what does this mean? ). If published, this will include your full peer review and any attached files.

**Do you want your identity to be public for this peer review?** For information about this choice, including consent withdrawal, please see our Privacy Policy .

Reviewer #1: No

Reviewer #2: No

---

## [Author Response · Author response to Decision Letter 1]

3 Feb 2025

PONE-D-24-43598

Perish the thawed? EDTA reduces DNA degradation during extraction from frozen tissue.

PLOS ONE

Shailender Kumar Verma, Ph.D.

Academic Editor

PLOS ONE

Dear Dr. Verma,

We thank you and reviewers 1 and 2 for providing insightful comments on our manuscript. As requested, we have revised the manuscript to address the concerns of each reviewer. We hope you agree that these changes have substantially improved the manuscript. In the following letter, you will find detailed responses to each reviewer’s questions. Author responses are in red typeface (in the pdf version, text formatting is not supported here) and identified with line numbers corresponding to the edits in the clean manuscript copy.

Reviewer #1: The manuscript addresses an important issue in DNA preservation during tissue thawing, offering a potential solution using EDTA. The study's relevance is significant for genomic and genetic research, especially in cryopreserved biological samples.

There are several questions.

1. Why was the focus placed specifically on EDTA and ethanol (EtOH)?

We have modified the text on lines 73-75 and 81-84 to explain why we focus specifically on EDTA and EtOH in this investigation.

Lines 73-75: We chose to evaluate EtOH as it is the most commonly used tissue preservative compatible with DNA preservation. It is widely employed for field collection and long-term storage of biological materials, particularly in natural history collections.

Lines 81-84: We chose to evaluate EDTA as it is a common ingredient in many preservative solutions (Nagy 2010) and is the primary active ingredient (Sharpe et al. 2020) in DESS (Seutin et al. 1991), a widely used tissue preservative that is as or more effective than EtOH in preventing DNA degradation in tissues from a wide range of organisms.

Were other preservatives considered but excluded, and if so, why?

We did not consider other preservatives. We agree that it would be very interesting to compare these two preservatives to others and don’t doubt that other preservatives may have similar or superior effects. It is simply beyond the scope of this investigation and the limits of our resources to compare EDTA and EtOH to a wider range of preservatives. However, because we compared the results of EDTA with those of EtOH, we can say that not every preservative will improve HMW DNA recovery from frozen tissue.

2. What novel insights does this study add beyond previous research on EDTA in DNA preservation?

Although freezing is typically considered the gold standard in DNA preservation, tissues must be thawed before DNA extraction. During the thawing period, even if brief, extensive DNA degradation can occur. The insight we provide is that 250 mM EDTA, pH 10, when applied to frozen tissue, is an easy, inexpensive, and effective way to mitigate this degradation.

3. Explain why authors chose these 10 organisms as target species.

We modified lines 111-116 to better explain our choice of species. Our aim was to test the efficacy of EDTA and EtOH for improving the recovery of HMW DNA from a broad range of taxa. We sampled specimens of 16 species selected to represent a wide range of animal taxa. These species included members of 16 genera, 12 families, and three phyla (S1 Table).

Lines 111-116: In this investigation, we sampled a wide range of species, storage conditions, and tissues to reflect a diverse selection of materials from which researchers may wish to isolate high molecular weight DNA. We chose 16 species that represent a wide range of animal taxa, including members of 16 genera, 12 families, and three phyla (S1 Table).

4. Were any controls implemented to account for potential variability in tissue type (muscle vs. visceral mass)?

We controlled for tissue type in every statistical model employed in this investigation by only comparing tissues of the same type and from the same species. For example, preservative-treated dorsal muscle samples from black seabass were compared only to untreated dorsal muscle tissue from black seabass, never to other tissue types or tissues from other species.

Could tissue composition impact the efficacy of EDTA treatment?

We agree with the reviewer that tissue composition likely impacts the efficacy of EDTA treatment. To acknowledge this, we added the following text:

Lines 446-449: Finally, we note that tissues of different types and taxonomic origin may vary widely in physical, chemical, and biological properties, e.g., hardness, surface area to volume ratio, permeability, pH, chemical composition, enzymatic activity, etc., and that these properties can influence the performance of preservative treatments.

To account for tissue-specific differences, we only made comparisons among tissues of the same type. However, it is encouraging that we observed significant improvement in HMW DNA recovery (%HMW, nY, or both) in frozen tissues treated with EDTA compared to untreated tissues in nine of 10 species examined, as we now state in lines 404-405. This indicates that EDTA treatment is effective across multiple tissues and storage conditions.

Lines 404-405: Indeed, we observed significant improvement in %HMW, nY, or both in nine of the 10 species examined.

5. Is there variability in tissue degradation across species that could confound the interpretation of EDTA's effectiveness?

We agree with the reviewer that species-specific differences in tissue degradation likely impact the effectiveness of EDTA. This is addressed in the text added on lines 446-449 (see above). To account for species-specific differences, we only made statistical comparisons among tissues from the same species. Because we tested each of the 10 species independently, differences among species do not impact our conclusion that EDTA treatment improves HMW DNA recovery from the frozen tissues of these species.

Would standardized storage conditions for all species improve the reliability of cross-species comparisons?

Standardized storage conditions would indeed improve the reliability of cross-species comparisons, and this would be an interesting subject for a follow-up study. However, in this investigation we tested EDTA treatment on tissues of each species independently. Our aim was to determine if this treatment was effective across many species, tissues and storage conditions rather than to compare the relative effectiveness among species, tissues and storage conditions.

6. What mechanisms might explain the species-specific variations observed in DNA preservation? Expand on potential biological or chemical factors.

We added a paragraph, lines 446-454 to address potential physical, chemical, and biological factors that might influence DNA preservation.

Lines 446-454 Finally, we note that tissues of different types and taxonomic origin may vary widely in physical, chemical, and biological properties, e.g., hardness, surface area to volume ratio, permeability, pH, chemical composition, enzymatic activity, etc., and that these properties can influence the performance of preservative treatments. Nonetheless, in this investigation, EDTA improved the recovery of HMW DNA across a wide variety of marine animal species and tissues, suggesting the potential of this method for broader application. Further investigation is warranted to determine if EDTA treatment can improve DNA recovery from frozen tissues of other marine and non-marine organisms across the tree of life, thereby adding value to the enormous variety of samples held in frozen research collections worldwide.

7. How does this study inform protocols for non-marine or plant tissues? Discuss the generalizability of findings to other types of biological samples.

We also address this question in the new paragraph on lines 446-454 (see above).

Reviewer #2: This study presents a comparative analysis of the effects of EDTA and ethanol on the preservation of high-molecular-weight (HMW) DNA in frozen tissues. The authors hypothesize that EDTA, known for its metal chelation properties, can reduce DNA degradation during the thawing process, which is a common step in DNA extraction that can lead to DNA damage. The study involved ten marine species, with tissues treated with EDTA at pH 10 or ethanol, and then analyzed for HMW DNA recovery. The results indicate that EDTA, particularly at pH 10, significantly improves the recovery of HMW DNA compared with that of ethanol and untreated tissues. These findings suggest that EDTA treatment can be a simple and effective method to increase DNA recovery from frozen tissues, which has implications for genetic and genomic research. Before this manuscript can be published, several questions need to be addressed:

1. Since EDTA is one of the primary active ingredients of DESS, why not compare it with DESS instead of choosing ethanol for comparison?

We have modified the text on lines 73-75 and 81-84 to explain why we focus specifically on EDTA and EtOH in this investigation.

Lines 73-75: We chose to evaluate EtOH as it is the most commonly used tissue preservative compatible with DNA preservation. It is widely employed for field collection and long-term storage of biological materials, particularly in natural history collections.

Lines 81-84: We chose to evaluate EDTA as it is a common ingredient in many preservative solutions (Nagy 2010) and is the primary active ingredient (Sharpe et al. 2020) in DESS (Seutin et al. 1991), a widely used tissue preservative that is as or more effective than EtOH in preventing DNA degradation in tissues from a wide range of organisms.

In our previously published work (Sharpe et al. 2020), we compared EDTA to DESS and each of the components of DESS individually and in pairwise combinations. That investigation showed that EDTA is the sole active ingredient in DESS. Moreover, DESS never outperformed EDTA and, in some cases, performed more poorly. Therefore, we concluded that comparing EDTA, the sole active ingredient in DESS, to DESS would not be as informative as comparing to EtOH, which acts by different chemical mechanisms.

2. In the Materials and methods section, the authors describe taking 100 mg of tissue and adding EDTA or ethanol, then placing it at 4°C for 12-24 hours before dividing it into 25 mg pieces for DNA extraction. Therefore, what is the purpose of the remaining three 25 mg tissues?

We modified lines 170-172 to clarify that one 25 mg subsample was removed from each 100 mg sample for DNA extraction. We chose 25 mg for DNA extraction because this is the optimal weight for the Qiagen DNEasy Blood and Tissue kit.

Lines 170-172: Subsequently, a 25 mg (avg. = 25.5 � 3.4 mg) subsample was removed from each 100 mg sample and transferred to the Qiagen DNeasy Blood and Tissue kit (Hilden, Germany) lysis solution for DNA extraction.

If only 25 mg is used for testing, would not it be better to directly immerse the 25 mg tissue into EDTA or ethanol for better protection of DNA? Please provide a detailed explanation.

We agree that a smaller (25 mg) piece of tissue would have a larger surface area to volume ratio and a shorter diffusion distance from its surface to its center, both of which might have made liquid preservation more effective. However, these same properties would also make the smaller samples thaw much more quickly, making it more likely that they might thaw accidentally before the preservative can be applied. The larger 100 mg samples are also easier to collect and handle. Since we aim to produce a practical method for most researchers, we decided that opting for 100 mg samples for treatment and 25 mg subsamples for DNA extraction was a reasonable tradeoff.

3. Regarding the above question, does the N=10 in Figure 1 refer to 10 marine fish or 10 marine invertebrate species, or is it 10 samples taken for testing?

We have modified Figure 1 and the associated figure legend to make the sampling scheme clearer.

Lines 140-145: Fig 1. Experimental Design. Three frozen tissue samples were collected from each of ten individuals of ten marine species (five marine fishes and five marine invertebrates) for a total of 300 samples. One sample from each individual was then thawed in EDTA (250 mM, pH 10) overnight at 4°C and the second was thawed in ethanol (95%) overnight at 4°C, after which DNA was extracted, analyzed, and compared to DNA extracted directly from the third frozen tissue sample, which did not receive liquid preservative treatment.

The S1 Table shows 10 fish, whereas the S2 Table shows 10 measurements for each treatment.

S1 Table describes the specimens used in this study, including 10 individuals of 10 species that were analyzed statistically, plus smaller numbers of individuals from six additional species.

S2 Table lists all measurements recorded for every sample analyzed in this study, including three samples for each of 10 individuals of 10 species that were analyzed statistically, plus smaller numbers of individuals from six additional species.

4. Lines 198-200: The authors calculated the concentration of HMW DNA for each sample by subtracting the low molecular weight DNA concentration from the total concentration of DNA present. However, quantitative results of the total DNA concentration and the low molecular weight DNA concentration are lacking.

We have added the requested data to S2 Table.

Please supplement more detailed data on how to calculate the concentration of HMW DNA.

Lines 200-224: We rewrote this section of the methods to better explain how we calculated %HMW and nY.

5. Lines 195-215 & S2 Table: Table S2 shows only the raw results of the Nanodrop 1000. The %HMW and nY values were calculated from the DNA concentration analyzed via the Agilent Technologies TapeStation 2200 DNA Analyzer. Please provide raw results from Agilent and the detailed data on how to calculate the concentration of HMW DNA.

We have added the requested data to S2 Table.

In addition, why are not Agilent's results provided in S2 Fig?

As stated in lines 359-362, we used gel electrophoresis in S2 Fig to reveal qualitatively similar trends for DNA extracts from tissues of six additional fish species. Because this is intended as a qualitative comparison, we chose not to present quantitative Tapestation data. However, we have added the Tapestation data to the revised S2 Table.

6. Lines 301--304: For four of the five invertebrate species (Amphioctopus sp., H. americanus, M. gigas, and P. vannamei), the %HMW values of DNA extracts from EtOH-thawed tissues were the same as or significantly lower than those from frozen tissues. However, for H. americanus, as shown in Fig. 2, the %HMW value from EtOH-thawed tissues was obviously higher than that from frozen tissues. Please clarify further.

Although the average value for %HMW is higher for EtOH than for fresh tissue of H. americanus in Figure 2, the difference is not statistically significant, as denoted by the same letters above the error bars. To clarify, we have changed the wording from “the same as” to “not significantly different from.”

Line 310-313: For four of the five invertebrate species (Amphioctopus sp., H. americanus, M. gigas, and P. vannamei), %HMW values of DNA extracts from EtOH-thawed tissues were not significantly different or significantly lower than those from frozen tissues.

7. Line 159: The samples were stored in preservative for 12 to 24 hours at 4°C. This means that the thawing times of the samples in the preservatives were inconsistent. Therefore, are the data in manuscript accurately compared under the same variable?

We used “12 to 24” hours as a generous estimate for “overnight.” We have changed all instances of “for 12 to 24 hours” to “overnight.” Overnight incubations are common in preservative studies as they typically provide more than enough time for preservative penetration and so, for practical purposes, are considered to be of the same length. Although we did not evaluate incubation time as a variable in this investigation, we plan to do so in a future investigation to determine a “minimum recommended” incubation time for this procedure.

8. Lines 147--151: Tissue samples were collected from different tissue parts of different species, such as the musculature posterior to

---

## [Decision Letter · Decision Letter 1]

12 Mar 2025

Perish the thawed? EDTA reduces DNA degradation during extraction from frozen tissue.

PONE-D-24-43598R1

Dear Dr. Distel,

We’re pleased to inform you that your manuscript has been judged scientifically suitable for publication and will be formally accepted for publication once it meets all outstanding technical requirements.

Kind regards,

Shailender Kumar Verma, Ph.D.

Academic Editor

PLOS ONE

Additional Editor Comments (optional):

Reviewers' comments:

Reviewer's Responses to Questions

**Comments to the Author**

1. If the authors have adequately addressed your comments raised in a previous round of review and you feel that this manuscript is now acceptable for publication, you may indicate that here to bypass the “Comments to the Author” section, enter your conflict of interest statement in the “Confidential to Editor” section, and submit your "Accept" recommendation.

Reviewer #1: All comments have been addressed

Reviewer #2: All comments have been addressed

2. Is the manuscript technically sound, and do the data support the conclusions?

Reviewer #1: Yes

Reviewer #2: Yes

3. Has the statistical analysis been performed appropriately and rigorously? 

Reviewer #1: Yes

Reviewer #2: Yes

4. Have the authors made all data underlying the findings in their manuscript fully available?

Reviewer #1: Yes

Reviewer #2: Yes

5. Is the manuscript presented in an intelligible fashion and written in standard English?

Reviewer #1: Yes

Reviewer #2: Yes

6. Review Comments to the Author

Reviewer #1: Authors have provided enough answers for my questions. I think that this manuscript could be accepted.

Reviewer #2: The authors have provided a detailed response to my comments, addressing each concern with clarity and justification. Their revisions to the manuscript are substantial and have significantly improved the overall quality and clarity of the research presented.

In their response, the authors have effectively addressed the concerns. For instance, they have provided a clear explanation for their choice of EDTA and ethanol (EtOH) as the focus of their study, emphasizing the practicality and relevance of these preservatives in the context of DNA preservation. They have also clarified the experimental design, particularly the use of 100 mg tissue samples for treatment and 25 mg subsamples for DNA extraction, which is a reasonable trade-off between practicality and effectiveness.

Overall, the revisions made to the manuscript have addressed the key issues, and the research presented is now more robust and clearly communicated. The study provides valuable insights into the use of EDTA as a preservative for improving HMW DNA recovery from frozen tissues, which has significant implications for genetic and genomic research.

7. PLOS authors have the option to publish the peer review history of their article (what does this mean? ). If published, this will include your full peer review and any attached files.

**Do you want your identity to be public for this peer review?** For information about this choice, including consent withdrawal, please see our Privacy Policy .

Reviewer #1: No

Reviewer #2: **Yes: ** Kaiyu Qian

---

## [Editor Report · Acceptance letter]

PONE-D-24-43598R1

PLOS ONE

Dear Dr. Distel,

I'm pleased to inform you that your manuscript has been deemed suitable for publication in PLOS ONE. Congratulations! Your manuscript is now being handed over to our production team.

Kind regards,

on behalf of

Dr. Shailender Kumar Verma

Academic Editor

PLOS ONE